# Person-Specific Template Matching Using a Dynamic Time Warping Step-Count Algorithm for Multiple Walking Activities

**DOI:** 10.3390/s23229061

**Published:** 2023-11-09

**Authors:** Valeria Filippou, Michael R. Backhouse, Anthony C. Redmond, David C. Wong

**Affiliations:** 1Institute of Medical and Biological Engineering, University of Leeds, Leeds LS2 9JT, UK; 2Warwick Clinical Trials Unit, University of Warwick, Coventry CV4 7AL, UK; michael.backhouse@warwick.ac.uk; 3Leeds Institute of Rheumatic and Musculoskeletal Medicine, University of Leeds, Leeds LS2 9JT, UK; a.redmond@leeds.ac.uk; 4Leeds Institute of Health Informatics, University of Leeds, Leeds LS2 9JT, UK; d.c.wong@leeds.ac.uk

**Keywords:** accelerometry, dynamic time warping, physical activity, step counting

## Abstract

This study aimed to develop and evaluate a new step-count algorithm, StepMatchDTWBA, for the accurate measurement of physical activity using wearable devices in both healthy and pathological populations. We conducted a study with 30 healthy volunteers wearing a wrist-worn MOX accelerometer (Maastricht Instruments, NL). The StepMatchDTWBA algorithm used dynamic time warping (DTW) barycentre averaging to create personalised templates for representative steps, accounting for individual walking variations. DTW was then used to measure the similarity between the template and accelerometer epoch. The StepMatchDTWBA algorithm had an average root-mean-square error of 2 steps for healthy gaits and 12 steps for simulated pathological gaits over a distance of about 10 m (GAITRite walkway) and one flight of stairs. It outperformed benchmark algorithms for the simulated pathological population, showcasing the potential for improved accuracy in personalised step counting for pathological populations. The StepMatchDTWBA algorithm represents a significant advancement in accurate step counting for both healthy and pathological populations. This development holds promise for creating more precise and personalised activity monitoring systems, benefiting various health and wellness applications.

## 1. Introduction

Being physically active is important for health and well-being as it helps to delay and prevent many chronic conditions [1]. Walking is one of the most common activities performed by humans in their everyday lives [2]. It has multiple benefits such as improving musculoskeletal, mental, metabolic, and cardiorespiratory health [3]. Walking and its characteristics have been previously measured in a laboratory environment using expensive and large systems [4,5] or in a free-living environment using inexpensive, time-consuming subjective measures such as surveys, diaries, and questionnaires, which demonstrate limited reliability [6,7].

Due to the increased demand for measuring physical activity in our everyday lives, several advances have been made in commercial wearable technology [5]. Accelerometers are at the core of the most popular wearables and are used to measure physical activity in terms of frequency, intensity, type, and duration [4]. Information from accelerometers can also be used to derive the number of steps taken during a time interval [8,9]. The total step count is increasingly being used as a metric to represent the overall health of individuals [5].

Although accelerometers offer several advantages, the algorithms used in consumer devices are usually targeted at a generally healthy population [10]. When they have been independently validated in pathological populations with impaired gait, the results have been mixed and there is evidence of poor accuracy in people with greater levels of impairment [11,12].

Our aim was to undertake a pilot study to develop and initially evaluate a new step-count algorithm that would perform well for individuals with pathological gait. Our approach uses person-specific templates of the accelerometer gait signal. We hypothesise that a personalised approach avoids issues that would be present in a model derived from a population with inter-individual variation. In the absence of a real pathological population, we used the simulated pathological gaits from healthy volunteers.

## 2. Related Works

Step-counting devices are commonly used to monitor physical activity levels, but many commercial and consumer-grade devices rely on “black-box” algorithms that cannot be replicated independently.

Studies have primarily focused on healthy populations, with few data available on step counting for individuals with chronic diseases or walking impairments. However, several studies have investigated the accuracy of step-counting devices in patient populations. For example, Marschollek et al. [13] tested four step-detection algorithms in both healthy and impaired participants and found that the algorithms yielded greater errors for impaired participants.

Capela et al. [14] developed a proprietary algorithm that achieved high accuracy in healthy participants, and Oudre et al. [15] developed a template-based matching algorithm to count steps in patient groups.

It is important to mention that among wearable devices, wrist-worn devices are the most acceptable to patients. Fortune et al. [16] and Genovese et al. [17] compared commercial devices with their own step-count algorithms and found significant differences in accuracy. Additionally, Huang et al. [18] developed their own algorithm integrated into a smartphone device, while several other studies used smartphones for their step-count algorithms.

Several different approaches have been applied to the development of step-count algorithms. This section provides an overview of the existing literature divided into three main categories: signal processing, template matching, and machine learning.

*Signal Processing:* Signal-processing approaches are relatively straightforward to implement. They often have low computational requirements, making them suitable for resource-constrained devices. They can filter noise and outliers using techniques such as moving average and median filters. The former is used to smooth variations in a dataset by calculating the average of a fixed window of consecutive data points and replacing the central point with this average value. The latter is similar to the moving average, but it replaces each data point with the median value of a window of neighbouring points. Dynamic thresholding methods can adapt to varying conditions and improve step detection accuracy. However, such algorithms may struggle to capture complex patterns in step data. They typically rely on predefined thresholds and heuristics, which may not generalise well across different individuals or activities. Their performance may be sensitive to noise and environmental factors. Manual tuning of parameters and thresholds is often required, making them less automated and adaptable.

Huang et al. [18] presented an orientation-free adaptive step detection (OFASD) algorithm for physical activity monitoring using a smartphone’s accelerometer. The OFASD algorithm uses three axes of acceleration and walking cycles to reduce false detections. It involves data preprocessing, a calibration training phase, and a step detection phase, requiring a training period for calibration. The algorithm allows the phone to be placed in any orientation, but it requires re-calibration if the position changes. Experimental validation with 10 participants wearing smartphones at different positions on the body at a constant walking speed showed no significant differences in step detection. The mean step detection accuracies ranged from 93% to 96%. Liu et al. conducted a statistical analysis of the duration of various activities, including the gait cycle. These findings could serve as a valuable reference for establishing predefined parameters for segmenting the walking acceleration signal [19].

*Template Matching:* Template-matching algorithms can capture the specific patterns and characteristics of steps. They can be effective in scenarios where step patterns are well defined and consistent. By using correlation coefficients, template-matching algorithms can quantify the similarity between templates and the input signal. On the other hand, such algorithms rely on predefined templates, which may limit their generalisability to different individuals, pathological groups, or activities. Generally, it can be time-consuming and subjective to manually extract templates from real accelerometer data. Based on the manual extraction, algorithms may struggle to handle variations in step patterns caused by different walking speeds or styles. The greedy nature of the algorithm may lead to sub-optimal step detection when overlapping steps occur.

In the template-matching category, Oudre et al. [15] introduced a step detection algorithm using accelerometer and gyrometer signals. The authors selected a library of 55 templates, including 14 for healthy subjects, 12 for orthopaedic subjects, and 29 for neurological subjects. The algorithm utilises the library of templates to recognise the start and end times of steps in the signal. It achieved a 98% recall and precision when tested on a database of 1020 recordings, including both healthy subjects and patients with neurological or orthopaedic issues. The process involves template comparison using correlation coefficients and a greedy process for step detection, with a post-processing step to discard false detections. Xu et al. [20] introduced a dynamic threshold estimation method for freezing-of-gait (FOG) detection. They employed an Artificial Neural Network (ANN) classifier in the data preprocessing stage to determine the optimal threshold value for each subject. Additionally, they proposed the sub-sequence dynamic time warping (sDTW) method to measure the similarity between sequences. Folgado et al. created a Python package called TSSEARCH, which provides a set of methods for sub-sequence search and similarity measurement in time series. This library was used for stride segmentation by providing a template and the whole signal. Using dynamic time warping, the authors identified which regions of the acceleration signal are more closely aligned with the template [21].

*Machine Learning:* In recent years, machine learning (ML) approaches have gained popularity as a way of estimating step counts. ML algorithms can learn complex patterns and adapt to differences between individuals. They have the potential to achieve high accuracy and robustness in step detection. ML algorithms can automatically extract relevant features from raw accelerometer data. They can generalise well to different individuals and activities, reducing the need for manual parameter tuning. However, ML algorithms often require a large amount of labelled training data to achieve good performance. They can be computationally intensive, especially deep learning models. Hyperparameter tuning and model selection can be challenging and time-consuming. The performance of ML algorithms heavily relies on the quality and representativeness of the training data.

Muñoz-Organero et al. [9] presented a three-phase algorithm that includes outlier detection, the generation of transition matrices capturing sequential patterns, and training an autoencoder to reconstruct acceleration data and identify individual steps. The experimental results of three participants showed that the algorithm achieved a recall of 88% and a precision of 50%. Small et al. [22] developed a step-count model combining machine learning and peak detection. They used an activity classification model trained with a self-supervised deep learning approach and the OxWalk dataset. The algorithm showed a mean absolute percent error of 12% and detection of 99% of actual steps. Luu et al. [23] explored different neural network architectures for sequence data modelling, including recurrent neural networks (RNNs) with long-short term memory (LSTM) or gradient reversal unit (GRU) cells and convolutional neural networks (CNNs) such as the WaveNet architecture. The hyperparameters of the models were selected based on extensive cross-validation on public datasets, achieving accuracies of 98%.

Overall, the literature contains a variety of techniques for step-count algorithms, ranging from signal-processing approaches to template-matching and machine learning methods. Each category has its strengths and limitations, and further research is needed to develop more accurate and robust algorithms for step-count estimation.

This paper introduces several significant contributions to the field of step counting. Our proposed step-count algorithm, StepMatchDTWBA, offers a simple and reliable method for tracking steps, catering to the needs of both individuals monitoring their physical activity levels and researchers studying gait patterns in clinical settings. This research aligns with the ongoing efforts to develop accurate step-counting algorithms for real-world scenarios, which holds crucial implications for public health and the promotion of physical activity. Throughout this study, we conducted activities of daily living, with a particular focus on investigating the impact of impairments.

The proposed StepMatchDTWBA algorithm offers a novel approach to step counting by leveraging dynamic time warping (DTW) barycentre averaging for creating personalised templates, significantly improving accuracy in both healthy and simulated pathological gait conditions. This innovation holds promise for enhanced step-counting precision, particularly in individuals with impairments, representing a substantial advancement in personalised activity monitoring.

## 3. Materials and Methods

This section describes the development of a new step-count algorithm (StepMatchDTWBA) based on template matching using the dynamic time warping (DTW) method and its extension, DTW barycentre averaging (DBA). The aim of StepMatchDTWBA is to enable more accurate step-count predictions in people with walking impairments. We first describe some of the constituent methods used and then describe the algorithm in full.

### 3.1. Background Methods

#### 3.1.1. Dynamic Time Warping (DTW)

DTW is an algorithm that calculates the optimal alignment of two sequences (signals) to determine their similarity. This is achieved by identifying flexible similarities in the time dimension (*x*-axis) by aligning the elements inside both sequences. The process results in a non-linear alignment that enables the matching of similar shapes, even when the sequences might be out of phase.

Specifically, we can visualise DTW via an n-by-m grid for two sequences of length n, m. A potential alignment between the components of the two sequences is represented by each point on the grid. An alignment pattern that matches every component of the two sequences is known as a warping path. There are several possible warping paths, but the shortest path is considered to be the best option [24].

The cost of the optimal alignment is computed recursively by:(1)$$D\left(Q_{n},P_{m}\right)=\delta \left(q_{n},p_{m}\right)+\min\left\{\begin{array}{c} D(Q_{n-1},P_{m-1})\\ D(Q_{n},P_{m-1})\\ D(Q_{n-1},P_{m}) \end{array}\right.$$
where $Q_{n}$ and $P_{m}$ are the sub-sequences $<q_{1},\ldots ,q_{n}>$ and $<p_{1},\ldots ,p_{m}>$, respectively. δ represents the Levenshtein distance between the elements of the sequences [25]. Once the path with the best alignment is identified, a similarity score, which describes the fit between the two sequences, is calculated as the total cost over the entire sequence length:(2)$$D\left(Q_{\left|Q\right|},P_{\left|P\right|}\right)=D\left(Q_{n},P_{m}\right)$$

#### 3.1.2. DTW Barycentre Averaging (DBA)

Barycentre averaging is a non-linear method of averaging a set of sequences [25]. DBA allows the resulting averaged sequence to keep the morphology of the individual sequences when they are out of phase, which may otherwise be averaged out in a simple Euclidean average.

DTW barycentre averaging (DBA) is an iterative algorithm in which DTW is used to align *N* sequences, $S=S_{1},\ldots ,S_{N}$, to an average sequence, $A=<a_{1},\ldots ,a_{T}>$.

An initial guess for *A*, $A^{\prime }=<a\prime_{1},\ldots ,a_{T}^{\prime }>$, is chosen randomly from *S*. $A^{\prime }$ is then updated by calculating the DTW similarity and path between $A^{\prime }$ and each individual sequence $S_{i}$. Each element of $A^{\prime }$ is updated as the barycentre of the elements associated with it during the previous step:(3)$$\begin{array}{c} A_{t}^{\prime }=barycentre\left(assoc\left(A_{t}\right)\right)\;\mathrm{where}\;barycentre\left(S_{1},\ldots ,S_{N}\right)=\frac{S_{1}+\cdots +S_{N}}{N} \end{array}$$

The assoc function is applied to the sequence $A_{t}$, which returns the associations between each coordinate in $A_{t}$ and the barycentric average sequence. These associations are then used in the barycentre calculation [25].

Figure 1 demonstrates several sequences that represent a single walking step and their DBA average. It shows how the overall shape of the sequences is preserved, despite the slight differences in the phase and frequency. Additionally, the figure provides a visual insight into the robustness of the DBA method in aligning and averaging time-series data, making it a powerful tool for analysing and comparing gait-cycle patterns.

### 3.2. Algorithm Development

The StepMatchDTWBA algorithm automatically detects the number of steps undertaken by a person, solely from wrist acceleration signals.

The key concept is to create a representative step from the acceleration signal as a template and then compare the template to the acceleration signal. It is assumed that a step corresponds to a match between the template and a particular area of the acceleration signal. The StepMatchDTWBA algorithm employs a template that models the overall shape of the signal, in contrast to many conventional step-count algorithms, which only take the number of peaks in a cyclical gait acceleration signal into account.

Our algorithm, StepMatchDTWBA, was inspired by several authors [20,26], but we mainly built upon the work of Micó-Amigo et al. [27], who created a personalised template of a walking period. However, there are two key differences in our approach. Firstly, our algorithm calculates the template length of a single step using both the peaks and troughs of the acceleration signal, whereas Micó-Amigo et al. only used the peaks. By considering both the peaks and troughs, our algorithm can capture the full range of motion involved in a step, making it more robust and accurate. Secondly, Micó-Amigo et al. [27] used correlation measures to identify the number of steps, whereas our algorithm uses dynamic time warping (DTW). DTW is a more powerful technique compared to correlation measures, especially when dealing with inconsistent gait morphology over time. By using DTW, our algorithm is more able to accurately identify the number of steps, even in cases where there is variability in the gait pattern.

All computations were undertaken in Python, and the source code has been uploaded in a GitHub repository that is available at https://github.com/ValeriaF22/Thesis-Project, accessed on 6 November 2023. However, the raw data are unavailable, as participants did not consent to share their data more widely.

### 3.3. Signal Preprocessing

To reduce the impact of high-frequency noise generated during data capture (caused, for instance, by muscle contraction), the accelerometer signal is first filtered using a 6th-order low-pass Butterworth filter with a 3 Hz cutoff frequency. The frequency of human activity is between 0 and 20 Hz, and almost all of the signal energy is contained below 3 Hz [28,29,30]. The flowchart in Figure 2 depicts the steps in the algorithm, with its output being the total number of steps. It is important to mention that the templates generated used the whole time series of the acceleration signal collected.

#### 3.3.1. Template Length Calculation

After the filtering process of the acceleration signal, the template length corresponding to a full gait cycle is extracted as follows:Obtain the unbiased autocorrelation signal of the input acceleration signal. This autocorrelation signal will be used to identify the periodicity of the signal, which is an important feature in detecting steps.
(4)$$\begin{array}{c} A_{\mathrm{unbiased}}=\frac{1}{N-|m|}\sum_{i=1}^{N-|m|}x_{i}x_{i+m} \end{array}$$
where *x${}_{i}$* is the acceleration signal, *N* is the length of the signal, and *m* is the time lag. An example output is shown in Figure 3, where the *x*-axis is the index for a given time lag *(m)*. By analysing this autocorrelation plot, the periodicity of the signal can be identified by measuring the time difference between the first two prominent peaks, which corresponds to its fundamental period.Identify the peaks of the autocorrelation signal, Pcorr, using scipy.signal.find_peaks. The peaks correspond to the periodicity of the input signal.Calculate the index (time) difference between the first two peaks of the autocorrelation signal ($min\_time=Pcorr_{2}-Pcorr_{1}$). This time difference provides an estimate of the minimum allowable time between neighbouring peaks in the input signal.Identify the peaks of the original acceleration signal using scipy.signal.find_peaks (Figure 4). The minimum allowable time between neighbouring peaks was set at κ×min_time. κ is an activity-specific parameter that was estimated experimentally (Table 1).Calculate the mean time difference between each consecutive pair of peaks in the original acceleration signal (from step 4). This value represents the average time period of the detected steps in the input signal.Repeat steps 3–5 for the troughs, which correspond to the negative peaks of the acceleration signal.Calculate the template length as the mean of the mean time difference of the peaks and the mean time difference of the troughs. This value represents the average time period of both the positive and negative peaks in the input signal.

#### 3.3.2. Template Signal Generation

The template length calculated in the previous step is used to generate the template signal. This length is used to segment the filtered acceleration signal into epochs, where each epoch has the same length and there is no overlap between epochs. We then use DBA to average the epochs together and create a single template signal for each acceleration signal, as demonstrated in Figure 1. The template is unique to each individual and activity.

#### 3.3.3. Template Matching to Calculate the Number of Steps Undertaken

The template signal can be used to estimate the step count for accelerometer data from the same individual. Data are first segmented into epochs of *length = template length*. Each epoch is then compared to the template signal using DTW, and the result is multiple similarity scores.
(5)similarityscores=dtw(template,epoch)

For each volunteer, several similarity scores are calculated by comparing the average template signal with each step signal sequence (epoch) using DTW. This similarity score is a measure of how closely the acceleration pattern of the epoch resembles the template.

To determine if a step has occurred, we introduce a threshold that acts as a cutoff point for similarity. A single value representing the upper bound of the confidence interval for the similarity scores was calculated for each activity under healthy and simulated pathological conditions. This value is used as a threshold to identify the number of steps. If a similarity score falls below this threshold, we consider it indicative of a step. The choice of using the upper bound of the confidence interval is grounded in statistics, as it provides a range that encompasses the true mean with a certain level of confidence. To determine the appropriate threshold values for healthy and simulated pathological conditions, we considered the inherent variability in the walking patterns of participants. For healthy individuals, we found that a threshold of “mean + 3 × std” provided the best results. This threshold takes into account slight variations in the walking patterns of healthy participants while maintaining the concept of similarity. On the other hand, pathological signals often exhibit greater variability due to irregularities, spikes, or fluctuations caused by the underlying condition. To handle this increased variability, we found that a more conservative threshold of “mean + 1 × std” was appropriate for simulated pathological conditions.
(6)threshold=mean+n×std
where n = 3 for healthy conditions and n = 1 for simulated pathological conditions.

By incorporating this upper bound of the confidence interval as the threshold, our algorithm can accurately identify and quantify steps, even when the sequences are not identical but exhibit similar patterns. This enhances the algorithm’s ability to capture variations in walking activity while ensuring reliable step counting, making it more adaptable to individual variations and differentiating between healthy and pathological cases.

## 4. Experiments

We investigated the proposed algorithm using data collected from 30 healthy participants. The recruitment of participants was conducted through email invitations and word of mouth by both staff and students at the University of Leeds. The eligibility criteria for participants included the ability to walk without discomfort for a duration of two minutes, the absence of any musculoskeletal or gait-affecting conditions, and an age of 18 years or older. Informed consent was obtained from every subject who took part in the research. Ethical approval for the study was granted by the University of Leeds (Ref#: MREC16-172).

### 4.1. Data Acquisition

Each participant wore a MOX tri-axial accelerometer (Maastricht Instruments, Maastricht, NL), which was held in place on the non-dominant wrist by an elasticated strap. The accelerometer had a measurement range of ±8 g and a sampling frequency of 100 Hz. The recorded signals were stored locally on the device and subsequently retrieved upon the completion of each participant’s trial.

The true number of steps was measured from video recordings. We recorded video of each participant as they walked using a smartphone camera. The camera was focused on the participant at a distance of approximately 2 m, such that the whole participant’s body was in the frame. A researcher (V.F.) used slow-motion playback of the videos to count steps and label the accelerometer data three times for reliability purposes [31].

Prior to attaching the activity monitor, participants received instructions and a video representation detailing the five activities they would engage in: a slow walk, a normal walk, and a fast walk along a GAITRite walkway spanning approximately 10 m, as well as ascending and descending a single flight of stairs, as demonstrated in Figure 5. Following the attachment of the accelerometer, participants were requested to execute a single jump to aid in aligning the video footage with the accelerometer data. Subsequently, participants undertook the activities in a sequential manner, with verbal reminders provided for each task. After completing the activities, participants were asked to perform another single jump.

Each sequence of tasks was executed twice: once in a state of normal health and subsequently under simulated pathological conditions. The decision to employ simulated pathological conditions was driven by the impracticality of recruiting actual patients, given the exploratory nature of the research, and the need to mitigate the potential risk of exposing an immunosuppressed population to SARS-CoV-2 during the pandemic. Under these simulated pathological conditions, participants were instructed to repeat the series of activities while adopting a shuffling gait and performing the tasks at a slower pace. A shuffling gait was defined as a situation in which the foot moved forward during initial contact or mid-swing, with the foot either remaining flat or striking with the heel, typically accompanied by shorter steps, reduced arm movement, and a forward-flexed posture [32]. They had the opportunity to practice before commencing data acquisition.

### 4.2. Model Validation

We used the root-mean-square error (RMSE) for each activity to measure the difference between the predicted and the true number of steps for all the participants. As a reminder, each activity was standardised to a distance of about 10 m (GAITRite walkway) in the case of walking and one flight of stairs for ascending and descending stairs.

The results were also assessed using Bland–Altman analysis. This first involved creating a Bland–Altman plot to visually inspect the agreement between the predicted and the true number of steps. The Bland–Altman plot is a scatter plot, in which the *Y*-axis demonstrates the difference between the two paired measurements, and the *X*-axis shows the average of these measures [33]. In our case, the *X*-axis shows the true number of steps instead of the average of the two methods [33,34].

Two key metrics can be derived from the Bland–Altman plot. The bias is used to identify any systematic difference between the predicted and the true number of steps. The limits of agreement (LOA) represent the range where most differences between the measurements of the two methods, gold standard and algorithm, will lie. They are calculated based on the mean and standard deviation of the differences between the measurements [35]. In this analysis, we used a V-shaped 95% LOA because we expected the total error in the predicted steps to increase with the total number of steps [36].

## 5. Results

### 5.1. Root-Mean-Square Error

The mean age of participants was 32.7 years (s.d. 12.7). Of the 30 participants, 14 identified as female. Their mean height was 171.5 cm (s.d. 7.1), and their mean weight was 69.2 kg (s.d. 13.6). The following results compare the algorithm developed for this study to several algorithms of different types: template matching [27], peak detection [37], and thresholding in the frequency domain [38].

Table 2 presents the comparative performance of StepMatchDTWBA and the template-matching algorithm across various activities, revealing StepMatchDTWBA’s superior results for slow and normal walking activities. However, for the remaining activities, the template-matching algorithm exhibited marginally better outcomes. To further illustrate the agreement between the predicted and the true number of steps for each algorithm, Bland–Altman plots, as shown in Figure 6, were generated, specifically focusing on walking at a slow speed under healthy conditions. The first plot demonstrates the agreement achieved using the template-matching algorithm, whereas the second plot represents the agreement obtained using StepMatchDTWBA. In this comparison, StepMatchDTWBA displayed a consistent bias of −0.59 (indicated by the red line) across different step counts, with the 95% limits of agreement (LOA) ranging from −3.48 to 2.30 (represented by the grey dotted lines). In contrast, the template-matching algorithm exhibited greater variability in the results, with a bias of 5.21 and 95% LOA ranging from −9.15 to 19.57.

Under simulated pathological conditions, the performance of all algorithms deteriorated compared to the healthy conditions. Table 2 reveals that the StepMatchDTWBA algorithm outperformed the *template-matching algorithm* across all five activities under these conditions. Figure 7 illustrates the increased error observed for both algorithms when operating under simulated pathological conditions, with a specific focus on slow-speed walking. StepMatchDTWBA demonstrated superior performance over the *template-matching algorithm*, exhibiting a bias of −3.48 (indicated by the red line) and 95% limits of agreement (LOA) ranging from −38.71 to 31.75 (represented by the grey dotted lines). Conversely, the *template-matching algorithm* displayed a bias of 56.72 and 95% LOA ranging from −64.90 to 178.35.

### 5.2. Testing the StepMatchDTWBA Algorithm on an External Public Dataset

To further validate the StepMatchDTWBA algorithm, we conducted a comprehensive test utilising the publicly available *Oxford Step-Counter* external dataset [39,40]. This dataset is unique, as it provides both raw accelerometer outputs and ground-truth values for step counts, offering an opportunity to assess the algorithm’s accuracy in real-world scenarios. Although the dataset features only two subjects, it remains a highly valuable resource for evaluating step-counting precision. The data collection process involved a Samsung S6 smartphone placed in six different positions: (1) hand, (2) front pocket, (3) back pocket, (4) neck pouch, (5) bag, and (6) armband. In our experiments, we focused solely on data obtained from the hand and armband locations. The Oxford Step-Counter dataset was originally collected by Salvi et al. [40], who also devised an algorithm based on windowed peak detection. Additionally, Pham et al. [41] developed another algorithm and tested it on the same dataset. Their approach incorporated peak detection and four essential features: minimal peak distance, minimal peak prominence, dynamic thresholding, and vibration elimination. Table 3 presents a comprehensive comparison of the true number of steps and the predicted number of steps generated by the StepMatchDTWBA and Pham et al. [41] algorithms. The StepMatchDTWBA algorithm demonstrated a maximum difference of four steps between the true and the predicted step counts, whereas the algorithm by Pham et al. exhibited a maximum difference of eight steps.

## 6. Discussion

Multiple approaches have previously been used to estimate step counts from accelerometers. Researchers have observed that there is a need for better step-count algorithms to be employed in (a) people with slow walking activity; (b) functionally compromised patient populations; and (c) when devices are worn on the wrist.

Our results align with those of prior research. We observed a reduced error (average RMSE of 1.91 steps over all activities) when using our step-count StepMatchDTWBA algorithm on data from normal gaits compared to data from simulated pathological gaits (average RMSE of 11.60 steps over all activities). We also conducted a comparative analysis between our algorithm and the nearest existing method, the *template-matching algorithm* [27], which also employs DTW. These two algorithms demonstrated similar performance when applied to healthy conditions. However, under simulated pathological conditions, our algorithm outperformed the *template-matching algorithm*.

The variation in algorithm performance can be attributed to various factors. Notably, the *template-matching algorithm* was originally designed for devices worn in different locations and targeted at distinct populations. Additionally, its primary objective was to calculate the step length rather than the total number of steps, with step counting serving as an intermediate stage.

The differences in error between the estimates of the healthy and simulated pathological gait steps can be attributed to various factors. In general, the acceleration signal recorded from the healthy condition group was periodic since the walking process is a rhythmic movement [42,43]. On the other hand, the acceleration signal recorded from the pathological condition group (in this case, simulated pathological) was less regular and contained more noise. The lack of regularity in the pathological group is one potential reason that existing methods showed a drop in performance. Additionally, the variability of the walking signal played an important role, especially for simulated pathological conditions. Healthy participants performed activities under both their normal routines and under pathological conditions. In their attempt to perform the activities under pathological conditions, the variability of their gait was much greater. The Bland–Altman plots showed that the range of the LOA for the healthy conditions was much smaller than the range of the LOA for the simulated pathological conditions.

The StepMatchDTWBA algorithm showed excellent performance when using the data from healthy participants. It also showed improved performance over existing algorithms when using the data from the simulated pathological gaits. This is very encouraging as it may enable more accurate step-count data estimates in patients who suffer from chronic impairments. This could be used to more accurately personalise treatment and recovery plans. Additionally, the algorithm was tested on data with varying speeds on flat surfaces and stairs. The results showed good performance in healthy participants (RMSE = 4.34 steps). On the other hand, the results were poorer for simulated pathological gaits (RMSE = 63.04 steps). In comparison to the *template-matching algorithm*, StepMatchDTWBA was almost five times more accurate. This might be because StepMatchDTWBA created more accurate templates since both peaks and troughs were considered.

Finally, the StepMatchDTWBA algorithm was also tested on the Oxford Step-Counter dataset. For the purposes of this study, only the accelerometer data from the hand and arm were used. The percentage error between the true number of steps and the predicted number of steps was ±1%. This suggests that the StepMatchDTWBA algorithm showed excellent generalisation to unseen external data from normal participants with normal gaits.

This study was limited in the following ways. First, the comparator algorithms by Micó-Amigo et al. [27], Aboy et al. [37], and Dirican et al. [38] did not have associated code and needed to be re-implemented. Second, the algorithms were tested only on healthy subjects, who performed the activities under healthy and pathological conditions. We hypothesise that our algorithm would perform worse on people with real walking impairments. Additionally, the algorithm was developed and tested offline. While we are unsure whether it is fast enough in real time, we note that the dynamic time warping used in DTW is of complexity $O(N^{2})$ and tends not to scale well. Lastly, the StepMatchDTWBA algorithm uses a threshold depending on the activity; therefore, activity classification should be performed prior to the use of this algorithm in order to select the appropriate threshold. The threshold can be updated as more data become available, increasing its generalisability and applicability.

The specific strengths of this study include the combination of two powerful techniques: DTW and DBA. The performance of our algorithm, StepMatchDTWBA, proved to be superior in both healthy and simulated pathological conditions in comparison to Micó-Amigo et al.’s algorithm, from which it drew inspiration. Additionally, the StepMatchDTWBA algorithm outperformed Aboy et al.’s baseline peak detection method and Dirican et al.’s thresholding method. Finally, the StepMatchDTWBA algorithm developed for this study is suitable not only for normal-speed walking but also for various walking speeds and climbing stairs. The majority of existing studies in the literature tested the step-count algorithms for normal walking, and some studies tested various walking speeds. However, only a few studies tested a step-count algorithm in more challenging activities such as climbing stairs, which represents a progression of previous work towards testing in real-life situations.

## 7. Conclusions

In this study, we compared a novel StepMatchDTWBA algorithm to an existing *template-matching algorithm* from the literature. The algorithms were tested using data collected from participants performing various activities under both healthy and simulated pathological conditions [44]. The StepMatchDTWBA algorithm demonstrated excellent accuracy for slow and normal walking speeds in the healthy condition group. For fast walking, stair ascent, and stair descent, the results were satisfactory and comparable to those of the *template-matching algorithm*. Although the StepMatchDTWBA algorithm’s performance in the simulated pathological condition group was not as strong as in the healthy group, it still outperformed the *template-matching algorithm*. These findings suggest the potential to develop more effective step-count algorithms for functionally impaired patient populations. This, in turn, could provide clinicians with more accurate and consistent results, accommodating individuals with varying functional abilities.

## Figures and Tables

**Figure 1 sensors-23-09061-f001:**
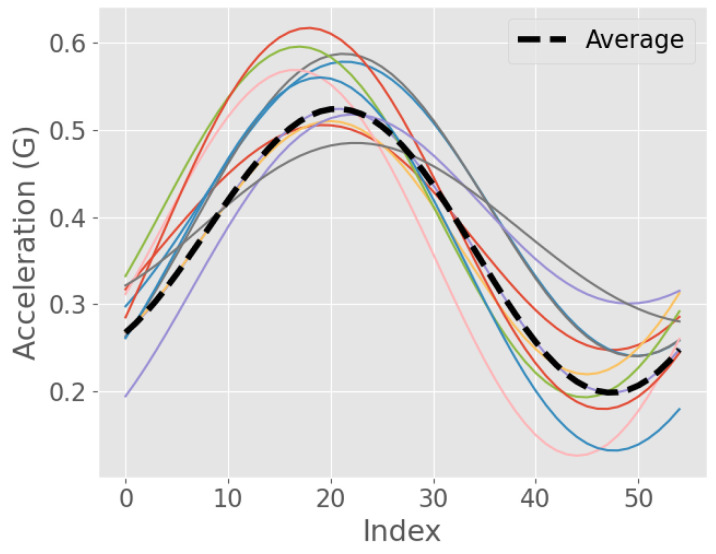
Averaging dynamic time warping of segmented acceleration signals in multiple colours using barycentre averaging.

**Figure 2 sensors-23-09061-f002:**
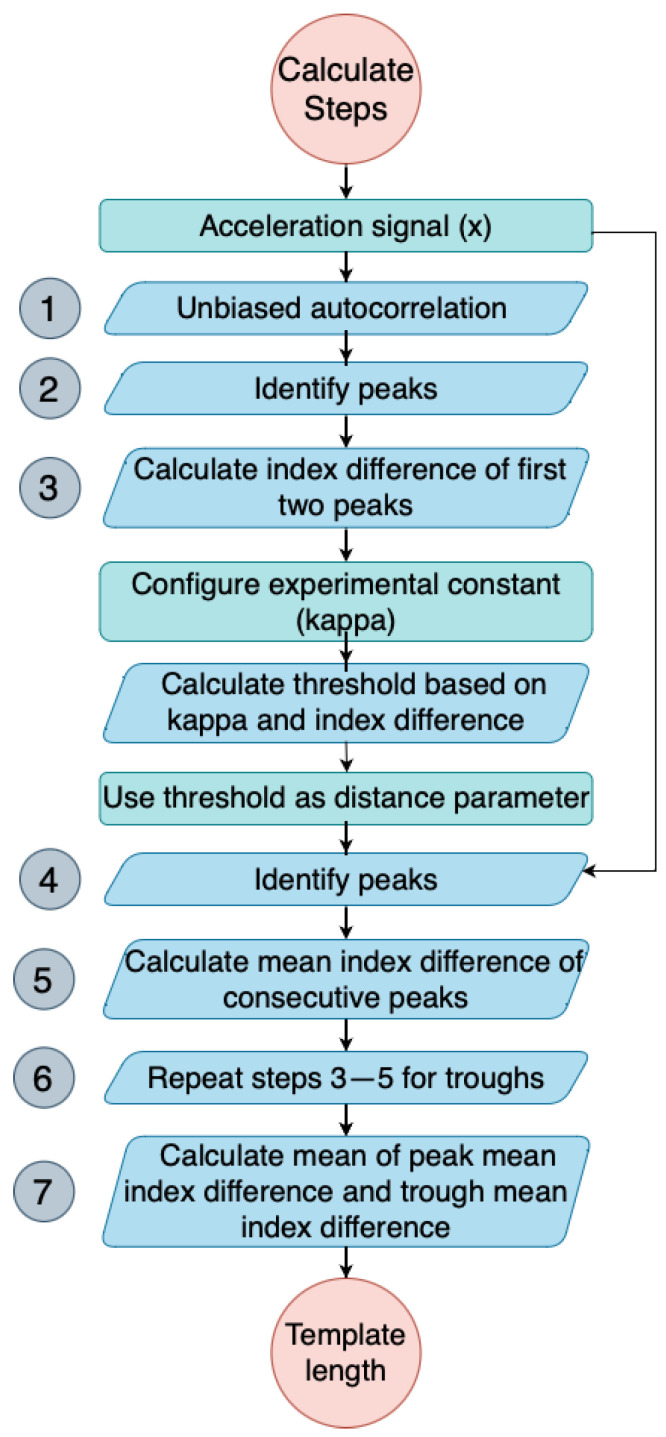
Flowchart of StepMatchDTWBA algorithm.

**Figure 3 sensors-23-09061-f003:**
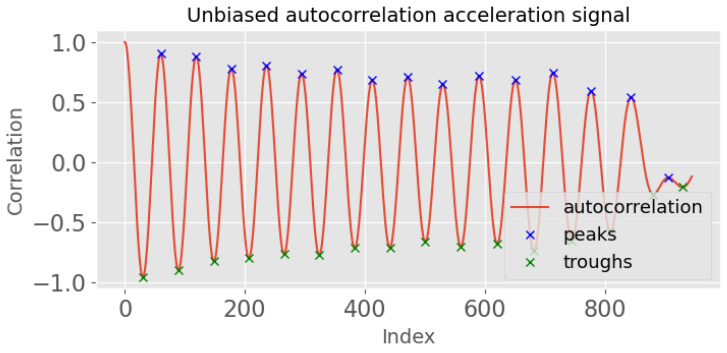
Unbiased autocorrelation signal of normal walking with peaks and troughs.

**Figure 4 sensors-23-09061-f004:**
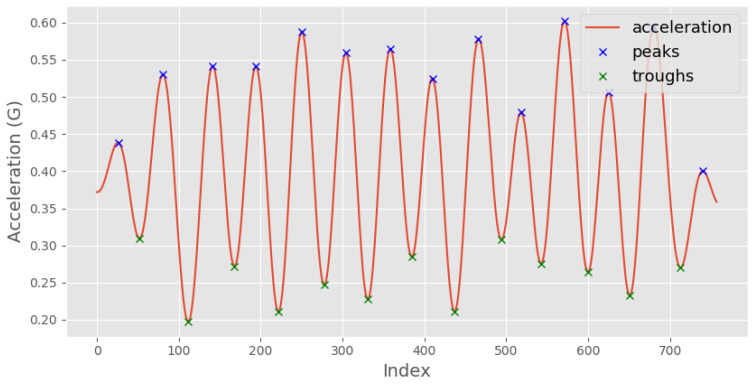
Acceleration signal of normal walking with peaks and troughs.

**Figure 5 sensors-23-09061-f005:**
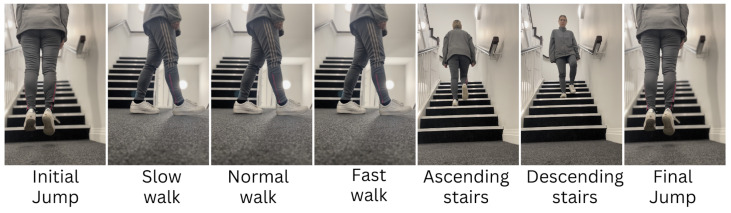
Sequence of activities performed by the volunteers during the experiment.

**Figure 6 sensors-23-09061-f006:**
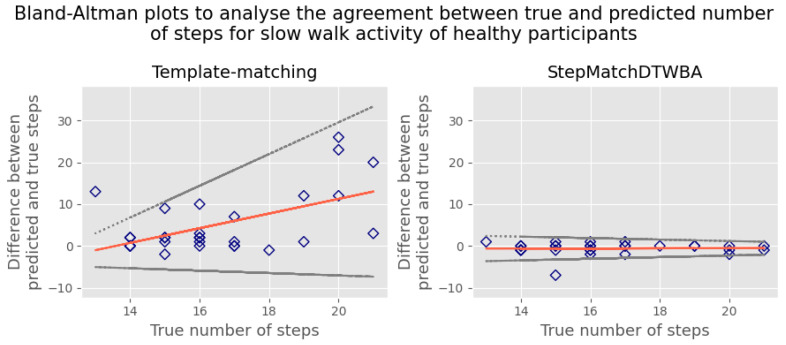
Bland–Altman plot under healthy conditions.

**Figure 7 sensors-23-09061-f007:**
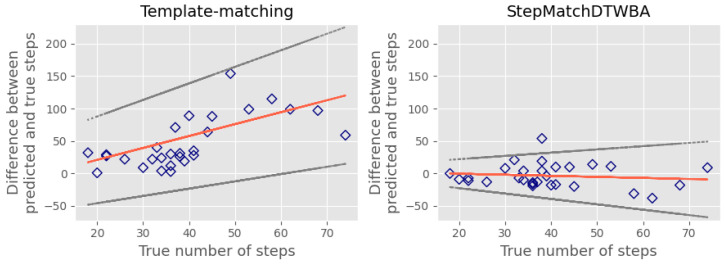
Bland–Altman plot under simulated pathological conditions.

**Table 1 sensors-23-09061-t001:** Constant, κ, used to calculate the adaptive threshold for each individual dynamic activity.

Activity	Normal	Simulated Pathological
Slow walk	0.65	0.95
Normal walk	0.10	0.90
Fast walk	0.10	0.75
Upstairs	0.10	1.30
Downstairs	0.10	1.30
All	0.10	1.20

**Table 2 sensors-23-09061-t002:** RMSE of steps per 10 m walking test results for the step-count algorithm for individual dynamic activities under healthy and simulated pathological conditions. Low and high confidence intervals are shown in brackets.

	Healthy				Simulated Pathological			
Activity	Benchmark Peak Detection	StepMatch DTWBA	Template Matching	Thresholding F-Domain *	Benchmark Peak Detection	StepMatch DTWBA	Template Matching	Thresholding F-Domain *
Slow walk	2.81	1.59	8.99	6.61	20.80	18.31	84.07	27.68
	(2.70, 2.92)	(1.39, 1.77)	(8.34, 9.59)	(6.34, 6.86)	(18.41, 22.94)	(17.22, 19.33)	(76.36, 91.14)	(65.55, 28.78)
Normal walk	4.03	1.00	1.35	4.09	15.94	14.04	50.87	23.13
	(3.91, 4.16)	(0.94, 1.06)	(1.28, 1.42)	(3.93, 4.23)	(13.76, 17.87)	(13.32, 14.72)	(47.03, 54.45)	(21.98, 24.22)
Fast walk	5.07	2.35	1.74	5.85	12.64	8.91	43.02	15.81
	(4.98, 5.16)	(2.25, 2.45)	(1.63, 1.84)	(5.74, 5.96)	(11.79, 13.44)	(8.47, 9.33)	(38.56, 47.06)	(14.74, 16.80)
Stairs ascent	4.55	2.04	1.90	5.37	10.01	7.98	42.12	38.06
	(4.41, 4.69)	(1.93, 2.15)	(1.76, 2.03)	(5.21, 5.52)	(9.43, 10.55)	(7.20, 8.70)	(39.18, 44.88)	(37.03, 39.07)
Stairs descent	5.16	2.57	1.11	6.36	9.07	8.74	59.22	34.63
	(4.98, 5.32)	(2.40, 2.74)	(1.08, 1.15)	(6.23, 6.48)	(8.27, 9.80)	(8.19, 9.25)	(56.15, 62.13)	(33.62, 35.61)
Average	4.32	1.91	3.02	5.66	13.69	11.60	55.86	27.86

* Frequency-domain.

**Table 3 sensors-23-09061-t003:** Number of steps calculated using StepMatchDTWBA and Pham et al. [41] algorithms on the Oxford Step-Counter dataset. Bolded items indicate the better-performing method for each position.

Position	True Steps	StepMatchDTWBA (Difference)	Pham et al. [41] (Difference)
Hand (1)	326	324 (−2)	323 (−3)
Hand (2)	340	338 (−2)	332 (−8)
Armband (1)	335	331 (−4)	335 (0)
Armband (2)	343	340 (−3)	335 (−5)

## Data Availability

Data are contained within the article.

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
