# Peer review of "Person-Specific Template Matching Using a Dynamic Time Warping Step-Count Algorithm for Multiple Walking Activities"

_sensors, 2023, doi:10.3390/s23229061_

Round 1

Reviewer 1 Report

Comments and Suggestions for Authors

This work is interesting.

1. Please distinguish the parameters and textual parts of the formulas rigorously. E.g., unbiased should be TEXT (\text).

2. Fig. 3: Please use the uniform structure. E.g. Experimental constant (kappa) -> Configure expe...

3. Fig. 6: no indications of the plot. E.g., what is the red curve? The dotted curve? I understand the Bland-Altman plot, but for a paper, you should provide all information for your plotted figures.

4. Please use the formula font ($$) for the parameters in the body text. E.g., L276.

5. Using different formats of decimals significantly lowers the scientific quality of the manuscript. E.g., you applied 95%, 93.4%, 

6. Please adhere a DATA AVAILABILITY STATEMENT. And the GitHub link is invalid, which can not be examined. It is a serious problem during reviewing.

7. The presentation is of good quality. Still, there exists minor flaws, such as:

L300: labelled

8. The author conducts some reviews of related work but ignores several recent documents that are particularly informative on the topics covered in this work.

For example, advanced DTW algorithms have already been used for personalized gait detection based on accelerometers (as well as gyroscopes), such as the open-source Time Series Subsequence Search Library published last year. Gaits were automatically detected according to the inquiry pattern, and the approach was validated on the CSL-SHARE human activity dataset.

Moreover, also using non-training, low-cost pure statistical algorithms, Rodrigues et al. proposed automatic activity/gait detection based on self-similarity matrices and novelty functions in 2022, including discovering cycles, anomalies, recurrences, etc, which is useful and efficient for automatic segmentation and gait counting: A long window can detect activity block changes, such as standing -> various kinds of walking; a medium time window can detect detailed activities during walking, such as going upstairs -> walking straight -> going downstairs; a short window can detect each gait.

The above two models or code bases can even been applied as SOTA peer methods to compare with your proposed algorithm.

In terms of gait statistics, SOTA includes How Long Are Various Types of Daily Activities? It analyzed 22 daily and sports activities, including dozens of walking-related, and concluded that adults' single motions (such as one gait) all last within about 1–2 seconds and are normally distributed among the population. An important reference for many related studies.

9. Coming from the mother tongue area, the authors have certainly no obvious problems with English expression, but the use of articles (a/the/no article) is often not rigorous.

Providing a repository link that cannot be opened is a huge waste of reviewer time...!

Comments on the Quality of English Language

See above.

Author Response

Dear Reviewer #1,

I would like to thank you for your review and insightful comments on my manuscript, " Person-specific template-matching using a dynamic time warping step count algorithm for multiple walking activities" Your feedback has been invaluable in improving the quality of my work, and I appreciate the time and effort you dedicated to the review process. You can find the updates in the document by looking at the red highlights.

I have addressed each of your points as follows:

Comment 1 from Reviewer #1:

Please distinguish the parameters and textual parts of the formulas rigorously. E.g., unbiased should be TEXT (\text).

Response to Comment 1:

Thank you for your feedback and the specific suggestion regarding the formatting of formulas in the manuscript. I appreciate your attention to detail and have taken your comment into account. You can find the updates in lines: 238, 270 and 288.

Comment 2 from Reviewer #1:

Fig. 3: Please use the uniform structure. E.g. Experimental constant (kappa) -> Configure expe...

Response to Comment 2:

The figure has been updated by starting with Configure.

Comment 3 from Reviewer #1:

Fig. 6: no indications of the plot. E.g., what is the red curve? The dotted curve? I understand the Bland-Altman plot, but for a paper, you should provide all information for your plotted figures.

Response to Comment 3:

Thank you for your feedback. I have added an explanation in the text for both red and grey lines.

“In this comparison, StepMatchDTWBA displayed a consistent bias of -0.59 \textcolor{red}{(indicated by the red line)} across different step counts, with the 95\% limits of agreement (LOA) ranging from -3.48 to 2.30 \textcolor{red}{(represented by the grey dotted lines)}.”

Comment 4 from Reviewer #1:

Please use the formula font ($$) for the parameters in the body text. E.g., L276

Response to Comment 4:

I have updated the formula font in the body text.

Comment 5 from Reviewer #1:

Using different formats of decimals significantly lowers the scientific quality of the manuscript. E.g., you applied 95%, 93.4%,

Response to Comment 5:

I have updated all the metrics to be a whole number, without any decimals.

Comment 6 from Reviewer #1:

Please adhere a DATA AVAILABILITY STATEMENT. And the GitHub link is invalid, which can not be examined. It is a serious problem during reviewing

Response to Comment 6:

Thank you for pointing that out. I have now changed the repository from private to public. Regarding the data availability, I have added the following sentence in the text in line 223.

“However, the raw data is unavailable, as participants did not consent to share their data more widely.”

Comment 7 from Reviewer #1:

The presentation is of good quality. Still, there exists minor flaws, such as: L300: labelled

Response to Comment 7:

I have updated the word labelled to labeled.

Comment 8 from Reviewer #1:

The author conducts some reviews of related work but ignores several recent documents that are particularly informative on the topics covered in this work.

For example, advanced DTW algorithms have already been used for personalized gait detection based on accelerometers (as well as gyroscopes), such as the open-source Time Series Subsequence Search Library published last year. Gaits were automatically detected according to the inquiry pattern, and the approach was validated on the CSL-SHARE human activity dataset.

Moreover, also using non-training, low-cost pure statistical algorithms, Rodrigues et al. proposed automatic activity/gait detection based on self-similarity matrices and novelty functions in 2022, including discovering cycles, anomalies, recurrences, etc, which is useful and efficient for automatic segmentation and gait counting: A long window can detect activity block changes, such as standing -> various kinds of walking; a medium time window can detect detailed activities during walking, such as going upstairs -> walking straight -> going downstairs; a short window can detect each gait.

The above two models or code bases can even been applied as SOTA peer methods to compare with your proposed algorithm.

In terms of gait statistics, SOTA includes How Long Are Various Types of Daily Activities? It analyzed 22 daily and sports activities, including dozens of walking-related, and concluded that adults' single motions (such as one gait) all last within about 1–2 seconds and are normally distributed among the population. An important reference for many related studies.

Response to Comment 8:

Thank you for mentioning these papers – I did not initially consider these as the analysis here precedes these works. I have read them and indeed they all seem very interesting. For the purpose of this paper, I have included the papers with title “TSSEARCH: Time Series Subsequence Search Library” and “How Long Are Various Types of Daily Activities? Statistical Analysis of a Multimodal Wearable Sensor-Based Human Activity Dataset” in my Related section. The following has been added to the document in line 85:

“Liu et al. conducted a statistical analysis on the duration of various activities, including the gait cycle. These findings could serve as valuable references for establishing predefined parameters for segmenting the walking acceleration signal \cite{Liu2022}.”

I believe TSSEARCH library is quite useful and the following has been added to the document in line 109:

“Folgado et al. created a python package called TSSEARCH, which provides a set of methods for sub-sequence search and similarity measurement in time series. This library was used for stride segmentation by providing a template and the whole signal. Using dynamic time warping, the authors identified which regions of the acceleration signal are more closely aligned to the template \cite{FOLGADO2022101049}.}”

The paper “Feature-Based Information Retrieval of Multimodal Biosignals with a Self-Similarity Matrix: Focus on Automatic Segmentation” is very interesting as well. From my understanding it would be more suitable in a paper that addresses activity classification rather than step count paper.

Comment 9 from Reviewer #1:

Coming from the mother tongue area, the authors have certainly no obvious problems with English expression, but the use of articles (a/the/no article) is often not rigorous.

Response to Comment 9:

We have checked the article, and we are satisfied that our use of articles is consistent with standard English grammar.

I believe these revisions have enhanced the manuscript, and I hope you find them satisfactory.

Reviewer 2 Report

Comments and Suggestions for Authors

Dear Authors,

thank you for sending the article titled: Person-specific template-matching using a dynamic time warping step count algorithm for multiple walking activities for the review process. The article seems interesting at first sight, but the authors should correct it according to below suggestions:

- please include the inclusion and exclusion criteria (participants)

-The authors used IMUs to obtain data. By using this kind of hardware we have a problem with drift. For example in the article titled: Gait recognition: A challenging Task for MEMS Signal Identification,  Smart Innovation, Systems and Technologies2019, KES-SDM 2019, pp.  473-484 authors obtain kinematic parameters. Unfortunatelly during experiment the drift was increase. IMU was corrected by using optical system (as a gold standard). Please describe in a few sentences how the authors dealt with this problem. Moreover, to improve article quality I suggest cite above manuscript and write a few sentences.

- Results. Please remove abbreviation (s.d). It should be write e.g. 32.7 (12.7) years. Every know that ii is std. dev.

- Table 3. Please include differencies between True, StepMatch and Pham.

Author Response

Dear Reviewer #2,

I would like to thank you for your review and insightful comments on my manuscript, " Person-specific template-matching using a dynamic time warping step count algorithm for multiple walking activities" Your feedback has been invaluable in improving the quality of my work, and I appreciate the time and effort you dedicated to the review process. You can find the updates in the document by looking at the blue highlights.

I have addressed each of your points as follows:

Comment 1 from Reviewer #2:

Please include the inclusion and exclusion criteria (participants)

Response to Comment 1:

This information already exists in the document. You can find the description of inclusion and exclusion criteria in line 297.

Comment 2 from Reviewer #2:

The authors used IMUs to obtain data. By using this kind of hardware we have a problem with drift. For example in the article titled: Gait recognition: A challenging Task for MEMS Signal Identification,  Smart Innovation, Systems and Technologies, 2019, KES-SDM 2019, pp.  473-484 authors obtain kinematic parameters. Unfortunatelly during experiment the drift was increase. IMU was corrected by using optical system (as a gold standard). Please describe in a few sentences how the authors dealt with this problem. Moreover, to improve article quality I suggest cite above manuscript and write a few sentences.

Response to Comment 2:

I appreciate your feedback and the opportunity to address the comment regarding IMU drift. I understand the concern, and I would like to provide clarification on this matter.

In response to the comment, it's important to note that Inertial Measurement Units (IMUs) typically consist of both accelerometers and gyroscopes. Gyroscopes are indeed more prone to drift over time, and I understand your point that IMU drift is generally associated with gyroscope measurements.

I wish to emphasize that our work primarily focuses on the analysis of accelerometer data, where drift is indeed less of a concern. Therefore, I believe there is less of a need to mention this issue since only accelerometers have been used.

https://www.sciencedirect.com/science/article/abs/pii/S0263224118308066

Comment 3 from Reviewer #2:

Results. Please remove abbreviation (s.d). It should be write e.g. 32.7 (12.7) years. Every know that ii is std. dev.

Response to Comment 3:

In line with standard reporting practices in the field, we have kept s.d. The inclusion of a label avoids any unnecessary ambiguity at the expense of two letters.

Comment 4 from Reviewer #2:

Table 3. Please include differencies between True, StepMatch and Pham.

Response to Comment 4:

I have included the difference between the True step count and both algorithms in Table 3, and highlighted the best performing method in each case.

I believe these revisions have enhanced the manuscript, and I hope you find them satisfactory.

Reviewer 3 Report

Comments and Suggestions for Authors

The collected activities should be included in the "4.1. Data acquisition" section as a series of continuous images to demonstrate how these actions are performed. Some of the Figures in the article do not clearly reflect the intended meaning, and any elements appearing in the images should be clearly labeled or described.

Comments on the Quality of English Language

The sentences in the manuscript need to be optimized.

Author Response

Dear Reviewer #3,

I would like to thank you for your review and insightful comments on my manuscript, “Person-specific template-matching using a dynamic time warping step count algorithm for multiple walking activities" Your feedback has been invaluable in improving the quality of my work, and I appreciate the time and effort you dedicated to the review process. You can find the updates in the document by looking at the blue highlights.

I have addressed each of your points as follows:

Comment 1 from Reviewer #3:

The collected activities should be included in the "4.1. Data acquisition" section as a series of continuous images to demonstrate how these actions are performed. Some of the Figures in the article do not clearly reflect the intended meaning, and any elements appearing in the images should be clearly labeled or described.

Response to Comment 1:

I appreciate your feedback regarding the figures and their description. I checked all of them and I have updated their description in the text. I removed figure 1 because I explain DTW in the text.

Regarding the new figure 1 I have added the following text in its description: “Additionally, the figure provides a visual insight into the robustness of the DBA method in aligning and averaging time-series data, making it a powerful tool for analysing and comparing gait cycle patterns.”

Regarding the new figure 2 I have added the following text in its description: “By analysing this autocorrelation plot, the periodicity of the signal can be identified by measuring the time difference between the first two prominent peaks, which corresponds to its fundamental period.”

I added a new figure at the Data acquisition section showing the sequence of activities performed by the volunteers.

Regarding figures 6 and 7 I have added an explanation in the text for both red and grey lines.

“In this comparison, StepMatchDTWBA displayed a consistent bias of -0.59 \textcolor{red}{(indicated by the red line)} across different step counts, with the 95\% limits of agreement (LOA) ranging from -3.48 to 2.30 \textcolor{red}{(represented by the grey dotted lines)}.”

Additionally, a figure at section 4.1 has been added, demonstrating the activities performed by the volunteers. 

Comment 2 from Reviewer #3:

The sentences in the manuscript need to be optimized.

Response to Comment 2:

We have checked the article, and we are satisfied that our sentences are consistent with standard English grammar.

I believe these revisions have enhanced the manuscript, and I hope you find them satisfactory.

Round 2

Reviewer 1 Report

Comments and Suggestions for Authors

I argue to accept this revised version.

Reviewer 3 Report

Comments and Suggestions for Authors

The reviewer is satisfied with the revisions by the authors and thus suggests acceptance of this manuscript for publication.

Comments on the Quality of English Language

There are basically no more problems.